



# Continuous increase in East Asia HFC-23 emissions inferred from high-frequency atmospheric observations from 2008 to 2019

Hyeri Park[1], Jooil Kim[2], Haklim Choi[3], Sohyeon Geum[1], Yeaseul Kim[3], Rona L. Thompson[4], Jens Mühle[2],
Peter K. Salameh[2], Christina M. Harth[2], Kieran M. Stanley[5], Simon O'Doherty[5], Paul J. Fraser[6], Peter G.
Simmonds[5], Paul B. Krummel[6], Ray F. Weiss[2], Ronald G. Prinn[7], and Sunyoung Park[1,3, *]

[1]Department of Oceanography, Kyungpook National University, Daegu, Republic of Korea
[2]Scripps Institution of Oceanography (SIO), University of California San Diego, La Jolla, California, USA
[3]Kyungpook Institute of Oceanography, Kyungpook National University, Daegu, Republic of Korea, Republic of Korea
[4]Norwegian Institute of Air Research (NILU), Kjeller, Norway
[5]School of Chemistry, University of Bristol, Bristol, UK
[6]Climate Science Centre, Commonwealth Scientific and Industrial Research Organisation (CSIRO) Oceans and Atmosphere,
Aspendale, Victoria, Australia
[7]Center for Global Change Science, Massachusetts Institute of Technology, Cambridge, MA, USA

*Correspondence to*: Sunyoung Park (sparky@knu.ac.kr)

**Abstract.** Trifluoromethane ($CHF_3$, HFC-23), one of the most potent greenhouse gases among HFCs, is mainly emitted to the
atmosphere as a by-product in the production of the ozone depleting legacy refrigerant and chemical feedstock
chlorodifluoromethane ($CHClF_2$, HCFC-22). A recent study on global HFC-23 emissions (Stanley et al., 2020) showed
significant discrepancies over 2014–2017 between the increase in the observation-derived (top-down) emissions and the 87 %
emission reduction expected from capture and destruction processes of HFC-23 at HCFC-22 production facilities mandated
by national phase-out plans (bottom-up). However, the actual sources of the increased emissions were not identified. Here, we
estimate the regional top-down emissions of HFC-23 for East Asia based on *in situ* measurements at Gosan, South Korea, and
show that the HFC-23 emissions from eastern China have increased from $5.0 \pm 0.4$ Gg yr$^{-1}$ in 2008 to $9.5 \pm 1.0$ Gg yr$^{-1}$ in 2019.
The continuous rise was contrary to the large emissions reduction reported since 2015 under the Chinese
hydrochlorofluorocarbons production phase-out management plan (HPPMP). The magnitude of the mismatch between top-
down and bottom-up estimates for 2015–2019 in eastern China was $\sim 23.7 \pm 3.6$ Gg, which accounts for $47 \pm 11$ % of the
global mismatch.
Given the location of HCFC-22 production plants in eastern China and the fraction of regional to global HCFC-22 production
capacities, the HFC-23 emissions rise in eastern China is most likely associated with known HCFC-22 production facilities
and thus, observed discrepancies between top-down and bottom-up emissions could be attributed to unsuccessful factory level
HFC-23 abatement and inaccurate quantification of emission reductions.



## 1 Introduction

Trifluoromethane ($CHF_3$, HFC-23) is mainly emitted as a by-product during the production of chlorodifluoromethane ($CHClF_2$, HCFC-22), a legacy ozone depleting refrigerant that is increasingly used as a chemical feedstock mainly for the manufacture of fluoropolymers (UNEP, 2018). HFC-23 has the highest 100-year global warming potential ($GWP_{100}$ of 12,690) among hydrofluorocarbons and a long atmospheric lifetime of ~228 years (WMO, 2018; Newman elt al., 2013). HFC-23 is included in the basket of greenhouse gases regulated under the 1997 Kyoto Protocol. Participating developed countries (Annex

I countries to the United Nations Framework Convention on Climate Change (UNFCCC)) are obligated to report their HFC-23 emissions to the UNFCCC. While developing countries (non-Annex I countries) are not obligated to report their HFC-23 emissions to the UNFCCC, some developing countries (e.g., China, India, South Korea, Mexico, and Argentina) participated in the UNFCCC's Clean Development Mechanism (CDM) between 2003 and 2014, a program to reduce HFC-23 emissions from HCFC-22 production facilities. Participating HCFC-22 facilities were required to report their HFC-23 reductions to the

UNFCCC and could trade Certified Emissions Reduction (CER) credits with developed countries.  During the CDM period, the total HFC-23 emission reductions provided by developing countries were compiled in 474 CDM monitoring reports (data available at https://cdm.unfccc.int/). However, note that not all HCFC-22 production facilities participated in the CDM program, and several facilities did not install HFC-23 abatement (destruction) technology (Montreal Protocol Technology and Economic Assessment Panel 2017 Assessment: hereinafter TEAP, 2017). Later, the HFC-23 by-production and emission

associated with the production of HCFC-22 have also been controlled by the Kigali Amendment to the Montreal Protocol on Substances that Deplete the Ozone Layer (MP), which was adopted in 2016 and entered into force in 2019 (Clark and Wagner, 2016) to preserve the climate benefit achieved by the MP.

   The global production of HCFC-22, and thus HFC-23 emissions, are heavily centered around East Asia. Notably, China, where more than 50 % of the global HCFC-22 (UNEP, 2018; Simmonds et al., 2018) has been produced since 2009, is

considered the largest HFC-23 emitter in the world. After the end of the CDM in 2014, the Chinese National Development and Reform Commission (NDRC) announced its plan for a complete reduction of all HFC-23 emissions from HCFC-22 production facilities by 2019 (NDRC, 2015). From 2014 to 2016, 13 HFC-23 abatement systems were installed at 15 HCFC-22 production lines that did not previously participate in the CDM program (TEAP, 2017). TEAP 2021 reported that the Chinese total of 32 HCFC-22 production lines in 2015 decreased to 18 lines in 2017, but also that the Chinese HCFC-22

production increased in 2017 compared to 2015. China has reported the annual HFC-23 emission reduction under the Chinese HCFC Production Phase-out Management Plan (HPPMP). The HFC-23 reductions reported by HCFC-22 production plants to the HPPMP were 45, 93, 98, and 99 % of total Chinese HFC-23 emissions in 2015, 2016, 2017, and 2018, respectively (UNEP, 2018; TEAP, 2021). According to China's HPPMP, the HFC-23 emissions from the whole of China should have been 7.5, 1.0, 0.3 and 0.3 Gg $yr^{-1}$ in 2015, 2016, 2017 and 2018, respectively.

Considering the activities for HFC-23 emissions reduction, we discuss (a) the 2008–2012 "CDM period", when mitigation of HFC-23 emissions had been introduced under the UNFCCC CDM to capture and incinerate HFC-23 co-production; (b) the





2013–2014 "end of CDM period" and (c) the 2015 to 2019 "post-CDM period or HPPMP period" when the Chinese HPPMP mitigation activities were underway (after CDM).

A recent top-down study based on atmospheric measurements showed that the global HFC-23 emissions (Stanley et al., 2020) had increased and in 2018 were higher than at any point in history. These results contradict the 87 % emission reduction anticipated over 2014–2017 from HFC-23 capture and destruction activities at HCFC-22 production facilities under the Chinese and Indian national phase-out plans initiated following the end of the CDM programs (UNEP, 2018; Say et al., 2019). The observed discrepancies between the top-down and the inventory-based "bottom-up" global HFC-23 emission estimates implied unexpectedly low abatement rates, unreported HCFC-22 production and resulting HFC-23 by-product emissions,

and/or unexpected substantial end-use and associated HFC-23 emissions. Therefore, these results at the global scale should be supported by investigations at the regional level where these HFC-23 emission controls have been implemented.

      In this study, we present top-down HFC-23 emissions in East Asia for 2008–2019, including for the first time, post-CDM HFC-23 emissions in this region. Our estimates combine long-term, high-frequency *in situ* observations of atmospheric HFC-23 concentrations made at a regional site of the Advanced Global Atmospheric Gases Experiment (AGAGE) network in East

Asia (Gosan station, Jeju Island, Korea, 33.3°N, 126.2°E) for 2008–2019, with a Bayesian atmospheric inverse method based on a Lagrangian particle dispersion model.

## 2 Method

### 2.1 *In situ* Observations at Gosan

      High-frequency atmospheric observations were made from 2008 to 2019 at Gosan (33.3° N, 126.2° E, 72 m a.s.l.) with a

custom build "Medusa" cryogenic trap and focusing system with Gas Chromatographic and Mass spectrometric (GC-MS) detection (Miller et al., 2008; Arnold et al., 2012). Measurements of ambient air samples, drawn from an air inlet at 72 m high (a.s.l.) every two hours, are bracketed with measurements of a working standard to correct for instrumental drift. The working standard is calibrated weekly against a secondary calibration tank prepared at the Scripps Institution of Oceanography (SIO). Measurements are reported on SIO calibration scales (e.g., SIO-07 scale for HFC-23 and SIO-05 scale for HCFC-22 and CFC-

11 in this study). The "Medusa" system is operated as part of the AGAGE program (Prinn et al., 2018). Precision (1-$\sigma$) derived from repeated analysis (*n*=12) of a working standard of ambient air was better than 1 % (i.e., the precision of HFC-23 < 1 %, HCFC-22 < 0.5 %, CFC-11 < 0.4 %) of background atmospheric concentrations for all compounds. The background concentrations (i.e., regional baseline) were determined using the AGAGE statistical pollution filtering algorithm (see more details in O'Doherty et al., 2001)

Gosan station is located on a remote cliff-top of the south-eastern tip of Jeju Island, south of the Korea Peninsula on the boundary between the Pacific Ocean and the Asian continent. The location is ideal for monitoring long-range transport from the surrounding regions with minimal local contamination. In the winter, the station often receives clean air masses characteristic of the "background" Northern Hemisphere directly from northern Siberia. In the summer, the monsoon brings



in southerly oceanic air masses characteristic of "background" in the Southern Hemisphere (Li et al., 2011). During other times,
the station samples polluted air masses from the Asian continent, particularly from China and South Korea. Some data gaps in
the measurement timeseries occurred in summer and early autumn due to typhoons and heavy rains. In particular, the data gap
from September 2016 to mid-April 2017 was due to impact of Super Typhoon Chiba and following reconstruction of the
station (Figure 1 and Figure 5).

## 2.2 Bayesian Inverse Modelling of Regional Emissions

### 2.2.1 Atmospheric model (FLEXPART)

We use the Lagrangian particle dispersion model FLEXPART version 10.4 (Pisso et al., 2019). It was used to model
atmospheric transport with meteorological fields from the Climate Forecast System Reanalysis (CFSR) model, at 0.5° x 0.5°
spatial and 1-hour temporal resolutions (Saha et al., 2010, 2011). 50,000 particles were released during a 30 min window
centered around each measurement time and tracked backwards in time for 20 days and from the measurement site throughout
the global domain. Particles were released from the sampling inlet height at Gosan site. Footprint sensitivities, which relate
the emissions at each 0.5° x 0.5° grid to measured concentrations at Gosan site, are estimated from the model outputs below
m in altitude. The total emission sensitivity was used to determine the variable size grids with grid cells ranging in size
from 0.5° x 0.5° to 12° x 12°, aggregated with larger grid size with decreasing sensitivity (For more details, refer to Kim et al.,
2021).

### 2.2.2 Inversion framework (FLEXINVERT+)

The FLEXINVERT+ Bayesian inverse model framework combines measured enhancement concentrations, air transport
estimated from FLEXPART, and *a priori* emissions to derive posteriori emissions. The model optimizes the posteriori
emissions to minimize the following cost function (Thompson and Stohl, 2014):

$$J(p) = 1/2(p-p^0)^T B^{-1}(p-p^0) + 1/2(H(p)-y)^T R^{-1}(H(p)-y) \tag{1}$$

Where p is the state vector of emissions, $p^0$ is *a priori* estimate vector, y is the measured enhancements, H is the matrices
derived from the FLEXPART backward simulations, B is the covariance matrix of *a priori* emission errors, and R is the
covariance matrix of measurement errors.

In the model framework, the "enhancement" concentrations were determined by linearly extrapolating the monthly mean
background concentrations and subtracting these extrapolated background concentrations from the measured concentrations
corresponding to each measurement time (Kim et al., 2021), and the bihourly enhancements data were used for the inversion
without temporal averaging.



*A priori* emissions of HFC-23 for East Asia and the global were determined as the 2008 *a priori* values of Stohl et al (2010) and Stanley et al (2020), respectively. The *a priori* emissions are 3.2 Gg yr$^{-1}$ for eastern China (8.5 Gg yr$^{-1}$ for entire China), 0.27 Gg yr$^{-1}$ for South Korea, 0.01 Gg yr$^{-1}$ for North Korea and 0.03 Gg yr$^{-1}$ Western Japan (0.08 Gg yr$^{-1}$ for entire Japan) (Stohl et al., 2010). The global *a priori* emission is 13.1 Gg yr$^{-1}$ (Stanley et al., 2020). Their magnitudes and spatial distributions were kept constant for the study period (2008–2019), ensuring that resulting trends of posteriori emissions are not biased by

any changes in *a priori* inputs and are informed solely by observations. More detailed information on *a priori* distributions, magnitudes, and uncertainties is given in the *SI appendix*.

The assigned total uncertainty of each enhancement ($\chi_{Measurement}$) was calculated as the square root of the quadratic sum of the three terms:

$$\chi_{Measurement} = \sqrt{(\chi_{Instrument})^2 + (\chi_{Background})^2 + (\chi_{Modelling})^2} \qquad (2)$$


Where, $\chi_{instrument}$ is instrumental precision of each compound, based on the repeatability of working standard measurements between sample observations (below 1 % for HFC-23); $\chi_{Background}$ represents an uncertainty in background concentrations, which was determined as one standard deviation of all the monthly means and extrapolated background concentrations over the observation period of 2008–2019 (about 0.2 ppt for HFC-23); and $\chi_{Modelling}$ is the model

representation uncertainty that is estimated as about 6 % of our monthly mean background concentrations.

We estimated HCFC-22 emissions using the same inverse framework as used for HFC-23. Optimal conditions of the inversion framework and its modelling performance were validated by analyzing CFC-11 emissions for eastern China based on Gosan CFC-11 data for 2008–2019 since the CFC-11 emissions were well-defined in recent studies (Rigby et al., 2019; Park et al., 2021) which used multiple inversion methods. See Supplement for details of the discussion.

**3 Result and discussion**

**3.1 Atmospheric mole fractions of HFC-23**

The time-series of atmospheric HFC-23 concentrations for 2008–2019 at Gosan is shown in Figure 1. The baseline values of the Gosan observations (i.e., background values representing regional clean condition without regional/local pollution events) are determined using a statistical method developed in AGAGE, which uses a 121-day moving window to identify positive

outliers from a Gaussian distribution that represent pollution events (O'Doherty et al., 2001). The annual averages of HFC-23 baseline concentrations at Gosan increased from 22.4 ± 0.4 ppt in 2008 to 33.3 ± 0.5 ppt in 2019 at a rate of 1.0 ± 0.1 ppt yr$^{-1}$. This rate is consistent with those from AGAGE stations Mace Head (53.3°N, 9.9°W) and Cape Grim (40.7°S, 144.7°E), which are representative of the global background monitoring stations in Northern Hemisphere and Southern Hemisphere, respectively. We observed pollution events at Gosan continuously throughout the record with significant enhancements above

background, which were influenced primarily by emissions from East Asian sources. Notably, the pollution signals were





significant even during the CDM period (for China, 2006 – early 2014) and later during the Chinese abatement activities under the HPPMP.

## 3.2 HFC-23 emissions from East Asia derived from atmospheric observations using inverse modelling

Unlike the global background monitoring stations, we observed large pollution events at Gosan continuously throughout the record, which has prompted a more detailed analysis of HFC-23 in East Asia to find and quantify potential source. Using the measurement enhancements in concentration units above the monthly-mean baseline, the atmospheric transport model (the Lagrangian particle dispersion model FLEXPART), and a regional inverse model framework (FLEXINVERT+), we calculate the qualitative spatial distribution of a posteriori emissions and estimate regional emissions of HFC-23 in East Asia including

eastern China, North Korea, South Korea and western Japan (Figure 2 and Table 1). These regions are based on the high sensitivity of the Gosan atmospheric measurements to HFC-23 emissions from the surrounding areas. The region denoted "eastern China" contains the nine provinces (Anhui, Beijing, Hebei, Jiangsu, Liaoning, Shandong, Shanghai, Tianjin, and Zhejiang) and "western Japan" contains the four regions (Chūgoku, Kansai, Kyūshū & Okinawa and Shikoku) (Rigby et al., 2019; Park et al., 2021; Kim et al., 2021; Western et al., 2022). A detailed description of the inverse model approach is given

in the Materials and Methods section below.

    Our top-down results show that HFC-23 emissions in eastern China account for 92–99 % of the total emissions in East Asia (Table 1). HFC-23 emissions in eastern China showed an overall increasing trend, with an appreciable decrease for 2010–2012 and an increase at the end of the CDM period (until 2014). During the HPPMP period, the eastern China HFC-23 emissions initially dropped (in 2015), but then increased continuously from $5.7 \pm 0.3$ Gg yr$^{-1}$ in 2015 to $9.5 \pm 1.0$ Gg yr$^{-1}$ in 2019. Our

results do not support the significant reduction in HFC-23 emissions that could be expected from the national mitigation activities under the Chinese HPPMP starting after the end of the CDM period. The cumulative total emissions were $21.9 \pm 2.1$ Gg (equivalent to $278.1 \pm 26.1$ MtCO$_2$-eq) for the CDM period of 2008–2012 vs. $28.2 \pm 3.6$ Gg (equivalent to $357.6 \pm 45.5$ MtCO$_2$-eq) for the post-CDM period of 2015–2019.The Chinese emissions of HFC-23 presented by previous studies agree with our estimates from 2008 to 2012, considering their large variability (Figure S1). Until this work, there was no HFC-23

emissions estimate for China and other East Asia countries for 2012 or later, including the post-CDM period. For South Korea, HFC-23 emissions increased from 0.0–0.2 Gg yr$^{-1}$ (min–max) in the CDM period to 0.1–0.5 Gg yr$^{-1}$ (min–max) in the post-CDM period. The distribution map of HFC-23 emissions (Figure 3) also showed this temporal change in South Korea's emissions. In 2019, the emissions from western Japan also increased slightly compared to previous years. The emissions from North Korea were near zero within modelling uncertainties. These national emission trends and magnitudes suggest that the

observed increase in East Asia emissions was predominantly driven by those from eastern China. The cumulative emissions from East Asia for the entire study period 2008–2019 were $73.8 \pm 9.2$ Gg which is equivalent to $999.6 \pm 117.3$ MtCO$_2$-eq.

    To gauge the inverse method's ability, we show the posteriori HFC-23 emission maps for 2008–2019, and compare them to known HCFC-22 factory locations taken from TEAP reports, company websites and media articles (Table S1) in Figure 3 and



Figure 4). The high emission flux densities were found predominantly in eastern China, where most of HCFC-22 factories are

located. The posteriori emission maps are used to examine how the spatial emission distribution has changed with the HFC-23 mitigation projects. We grouped a posteriori HFC-23 emission maps for the CDM period, the end of the CDM period, and the HPPMP period (see Figure 3). Over the three periods, the broad patterns in the emission hotspots did not change much, except for the end of the CDM, when we detected relatively stronger emission signals from eastern China, and a slight increase in the overall emissions. As revealed in the HFC-23 emission time series in Figure 2, the emissions during the HPPMP period

were similar to or even higher than those during other periods.

### 3.3 Estimation and comparison of HCFC-22 emissions in eastern China

To confirm the link of HFC-23 emissions in eastern China to HCFC-22 production, we estimated HCFC-22 emissions from eastern China in the same inverse framework used for HFC-23 (Figure 6(a)).

High-frequency HCFC-22 observation data for 2008-2019 show persistent pollution events, clearly implying that HCFC-22 emissions have been emanating from the surrounding regions, while the regional baseline concentrations of HCFC-22 show a similar increasing trend as the global NH baseline. It should be noted that HCFC-22 baseline concentrations at Gosan drop periodically in summer due to strong intrusion of SH tropical air masses with low HCFC-22 concentrations during the East Asia summer monsoon (Li et al., 2018).

The continuing rise in the emissions seems to imply that HCFC-22 production and consumption in the region have been increasing and the contribution of dispersive use is still significant, although its production for dispersive use is currently being phased out in developing countries by the Montreal Protocol. HCFC-22 emissions from the whole of China were inferred from the faction of the population in eastern China. Results are consistent not only with previous studies but also with the invertory-based HCFC-22 emissions as shown in Figure 6(b), suggesting that population density still serves as good proxy for HCFC-

22 emissions, and that the bottom-up emissions for the whole of China are relatively well-defined.

### 3.4 Estimation of HCFC-22 production in eastern China

HCFC-22 production data for entire China from 2008 to 2018 are given in the TEAP 2021 report, but only for information in 2015 and 2018 for eastern China. To estimate the missing HCFC-22 production data for eastern China, we calculated the eastern China production fractions for other years (green dashed line in Figure 7), which were derived based on the following

equation (3) using the 2015 and 2018 fractions. We assumed the eastern China production fractions were correlated exponentially with time in the following form.

$P_{22}(E. Ch)/P_{22}(Ch) = A*exp[k*(year-2008)]$                                           (3)



Here $P_{22}(E. Ch)$ and $P_{22}(Ch)$ are HCFC-22 productions from eastern China and the entire China, respectively. $A$ is the initial fraction in 2008. $k$ is a continuous growth rate. They were determined from the known HCFC-22 production fractions for the years of 2015 and 2018. Then we can calculate eastern China fractions of HCFC-22 productions for other years.

  This analysis was based on the assumptions that the eleven HCFC-22 facilities reported to UNEP are all the plants producing HCFC-22 in entire China, and that their production reports are correct. If un-reported and/or newly built plants exist at

unknown locations, actual fractions of HCFC-22 productions for eastern China could be changed, but our estimation of the abated HFC-23 emissions for eastern China in main text would not be affected because the emissions abatement action under both the CDM and HPPMP programs must have been taken in the reported facilities. It is also assumed that a varying rate between the 2015 and 2018 production fractions in eastern China can be applied consistently over time. The annual HCFC-22 productions in eastern China calculated from the inferred eastern China fractions (green solid line in Figure 7) accounted for

about 60–90 % of the annual totals for China during 2008–2018.

## 3.5 Discrepancy between bottom-up and top-down eastern China estimates

  Inventory-based bottom-up estimates of HFC-23 emissions can be derived from reports of HCFC-22 production multiplied by yearly emissions factors of co-produced HFC-23 (i.e., the total mass of emitted HFC-23 as a fraction of total HCFC-22 production) (TEAP, 2021), and from national emissions reported to the UNFCCC for Annex-I countries. The former is

important for non-Annex I countries that do not report emissions to the UNFCCC and refers to the expected emissions when no abatement activities would be applied. The historical HCFC-22 production data of entire China for 2008-2018 are given in the TEAP 2021 report (Figure 7), but not for our inversion domain of eastern China. To generate unabated bottom-up HFC-23 emissions estimates in eastern China (red dotted line in Figure 8), we used yearly varying emission factors of co-produced HFC-23 in the range of 2.32–2.78 % (Stanley et al., 2020; TEAP, 2017; TEAP, 2021; UNEP, 2017), and the annual HCFC-22

production in eastern China was calculated using equation (3) in Section 3.4, which expressed an exponential correlation of the eastern China production fractions with time.

  HFC-23 emissions for eastern China estimated with abatement (red dashed line in Figure 8) include emissions reduction expected from the CDM and the HPPMP programs reports (UNEP, 2018). The anticipated reductions for eastern China were scaled down by the fractions of HCFC-22 production capacities in eastern China determined by equation (3) in Section 3.4.

The CDM abatement emission of HFC-23 in eastern China was ~8.2 Gg in 2012 and decreased close to zero by the end of 2014. During the HPPMP activities, we estimate that the HFC-23 emissions reductions of eastern China by the HPPMP should have been 4.9, 11.0, 13.4 and 12.8 Gg yr$^{-1}$ in 2015, 2016, 2017 and 2018, respectively, based on reported abatement rates (UNEP 2018; TEAP, 2021). Therefore, the deduced bottom-up emissions with abatement for eastern China (red dashed line in Figure 8; note that the 2019 bottom-up emission was assumed constant after 2018 because the 2019 reduction of HFC-23 was not reported yet) dramatically dropped in 2015 and remained very low since 2016, contrary to the increasing emissions

shown in our observation-based top-down emissions estimates (red solid line and shading in Figure 8).



It is interesting that the abated bottom-up estimates for 2012–2014 were larger than our observation-based emissions, although the global emissions in 2013–2014 also showed a very similar anomaly. Possible explanations are: (a) excess reduction above the reported abatement capacity might occur and/or (b) branches of HCFC-22 manufacturing facilities might be installed and operated at other locations outside eastern China, and their production capacities were accounted for and recorded as those of main (headquarter) factories located in eastern China (Figure 4). If the latter is the case, then our eastern China HCFC-22 production fractions could be overestimated, thereby generating overestimates of no-abatement/abatement bottom-up emissions.

Despite these open questions, our observation-based top-down emissions were much larger than those expected from abatement/reduction activities from 2015, and it is clear that abated emissions in eastern China were supposed to be near zero since 2015 under the reduction plan. The total excess emission (i.e., the integrated difference between our top-down emissions and the bottom-up abatement emissions) of eastern China for 2015–2019 is $23.7 \pm 3.6$ Gg (equivalent to around $301.3 \pm 45.5$ MtCO$_2$-eq) (yellow + red hatched area of Figure 8), which represents ~3 % of the total greenhouse gases emissions of 9,876.5 Mt-CO$_2$ for the whole of China for 2019 (Greenhouse Gas Emissions by Country, 2022).

## 4 Conclusion

Our estimated HFC-23 emissions from East Asia (Figure S2) accounted for $49 \pm 11$ % of the global top-down emissions between 2008–2018 taken from Stanley et al (2020). Since our results show HFC-23 emissions from eastern China account for most of the total emissions for East Asia (92–99 %), we focus here on how eastern China has influenced global emissions trends (as shown in Figure 8).

The expected reduction in HFC-23 emissions from eastern China for 2015–2017 accounts for 78–82 % of the reported global HFC-23 emission reductions. The bottom-up emissions with abatement for eastern China demonstrate a similar temporal trend shown in the corresponding global values. The cumulative total excess emissions in eastern China of $7.9 \pm 1.8$ Gg for 2015–2017 (top-down minus bottom-up; yellow hatched area of Figure 8) account for 47 % of the global cumulative excess emissions (gray hatched area of Figure 8). Given that the 2015–2019 excess emission in eastern China is $23.7 \pm 3.6$ Gg, with an assumption that 47 % of the global excess in eastern China would be maintained for 2018–2019, the global cumulative excess emissions are expected to reach to $50.5 \pm 7.6$ Gg (around $641.1 \pm 96.8$ MtCO$_2$-eq) for 2015–2019. This means that if the HFC-23 emissions reduction reported by China and other countries had actually taken place, unknown HCFC-22 production of as much as $2079 \pm 314$ Gg for 2015–2019 would be required to explain the global excess HFC-23 emissions of $50.5 \pm 7.6$ Gg. Approximately 400 Gg per year of unknown HCFC-22 was produced during 2015–2019, which is more than two-thirds of the total HCFC-22 production per year in entire China (Figure 7). The eastern China excess HFC-23 emission of $23.7 \pm 3.6$ Gg should result in unknown HCFC-22 production of $977 \pm 147$ Gg, which is equivalent to 136–184 % of the total quantity of HCFC-22 produced in eastern China or 118–159 % of the total production in A5 parties reported for 2018.

On the other hand, while re-considering the eastern China HFC-23 emissions in terms of HCFC-22 production, we estimated HCFC-22 emissions from eastern China by using the HCFC-22 measurement data obtained at Gosan for 2008–2019 (Figure



290    5) in the same inverse frameworks used for HFC-23 (Figure 6(a)), and then inferred the entire China emissions of HCFC-22 from the fraction of the population of eastern China. These observation-based China emissions of HCFC-22 were consistent not only with previous top-down studies but also with the inventory-based HCFC-22 emissions as shown in Figure 6(b), thereby suggesting that the bottom-up emissions for China were relatively well-defined and thus the associated HCFC-22 production records were reasonably accurate. Therefore, using the fraction of the HCFC-22 production in eastern China against

295    the global total production (TEAP, 2021; Figure 7), we calculated the eastern China HFC-23 emissions by scaling down the global top-down HFC-23 emissions given in Stanley et al. (2020). The HCFC-22 production-scaled HFC-23 emissions (gray lines in Figure 9) agree well with our observation-inferred emissions for eastern China even during the HPPMP period, suggesting that recent increases in both global and regional HFC-23 emissions is predominantly associated with known HCFC-22 production facilities. So, observed discrepancies in top-down and bottom-up HFC-23 emissions estimates are more likely

300    due to unsuccessful reduction processes of HFC-23 at factory level and inaccurate quantification of emission reductions, rather than the existence of unreported, unknown HCFC-22 production. Overall, our finding of such unexpected regional emissions of HFC-23 can help identify any responsible industrial practices and make up more effective regulatory management to avoid the unwanted emissions.



**Data availability**

Data from Gosan, Mace Head and Cape Grim stations are available from the AGAGE website (http://agage.mit.edu/data/agage-data) or upon request by contacting SP, SOD, and PBK.

**Author contribution**

HP, SG, YK and SP carried out the Gosan measurements. HP analyzed the data with SP and performed inverse modelling with contributions from JK, RLT and HC. JM, PKS, CMH and RFW supported the calibration, precision and data quality of the long-term observations at Gosan. SOD, PJF, PGS and PBK provided the *in situ* measurement data from the global baseline stations. RFW and RGP supported the AGAGE network. HP and SP wrote the manuscript with contributions from all co-authors.

**Competing interests**

**Acknowledgements**

This research was supported by the National Research Foundation of Korea (NRF) grant funded by the Korean government (MSIT) (no. 2020R1A2C3003774). The NASA Upper Atmosphere Research Program supports AGAGE (including partial support of Mace Head and Cape Grim) through grant NNX16AC98G to MIT, and grants NNX16AC96G and NNX16AC97G to SIO and multiple preceding grants. Mace Head station is supported by the UK Department of Business, Energy and Industrial Strategy (BEIS contract 1537/06/2018). Cape Grim station is primarily supported by the Australian Bureau of Meteorology and CSIRO, with additional support from the Australian Department of Agriculture, Water and the Environment (DAWE) and Refrigerant Reclaim Australia (RRA).

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



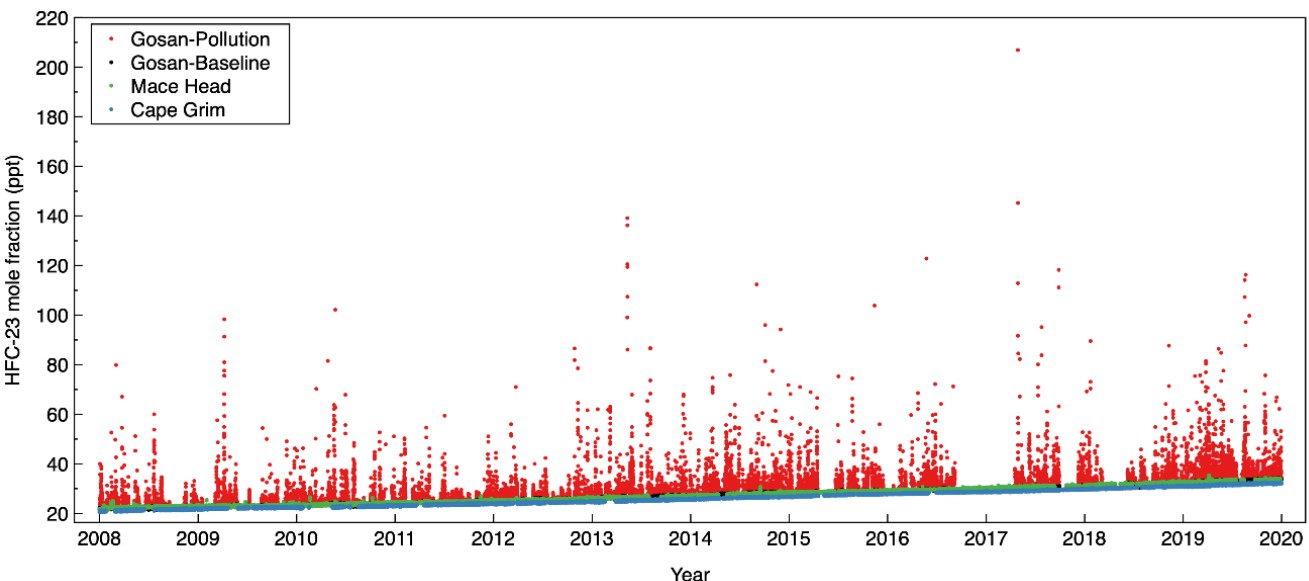

**Figure 1: Atmospheric HFC-23 concentrations observed from 2008 to 2019 at Gosan. The green and blue dots show HFC-23 concentrations observed at Mace Head and Cape Grim, respectively, for comparison.**





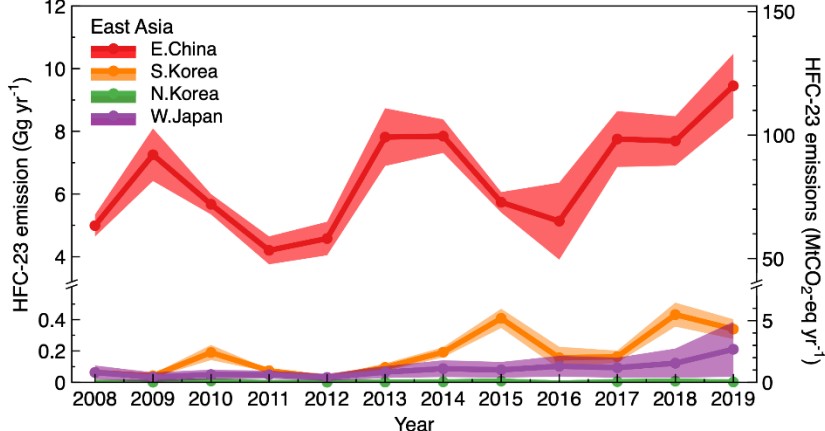

**Figure 2:** HFC-23 emissions estimate in East Asia using the FLEXINVERT+ inverse method and in-situ HFC-23 measurements at Gosan, Jeju Island, South Korea. Top-down emissions estimates are shown for eastern China (red), South Korea (orange), North Korea (green), and western Japan (purple) for the years 2008–2019. Each line and the shading denote the a posteriori emission mean of the results from 18 different *a priori* settings and the 2-σ uncertainty, respectively. Right-hand axis is scaled by the 100-yr GWP for HFC-23 to show emission magnitudes in MtCO₂-eq yr$^{-1}$.



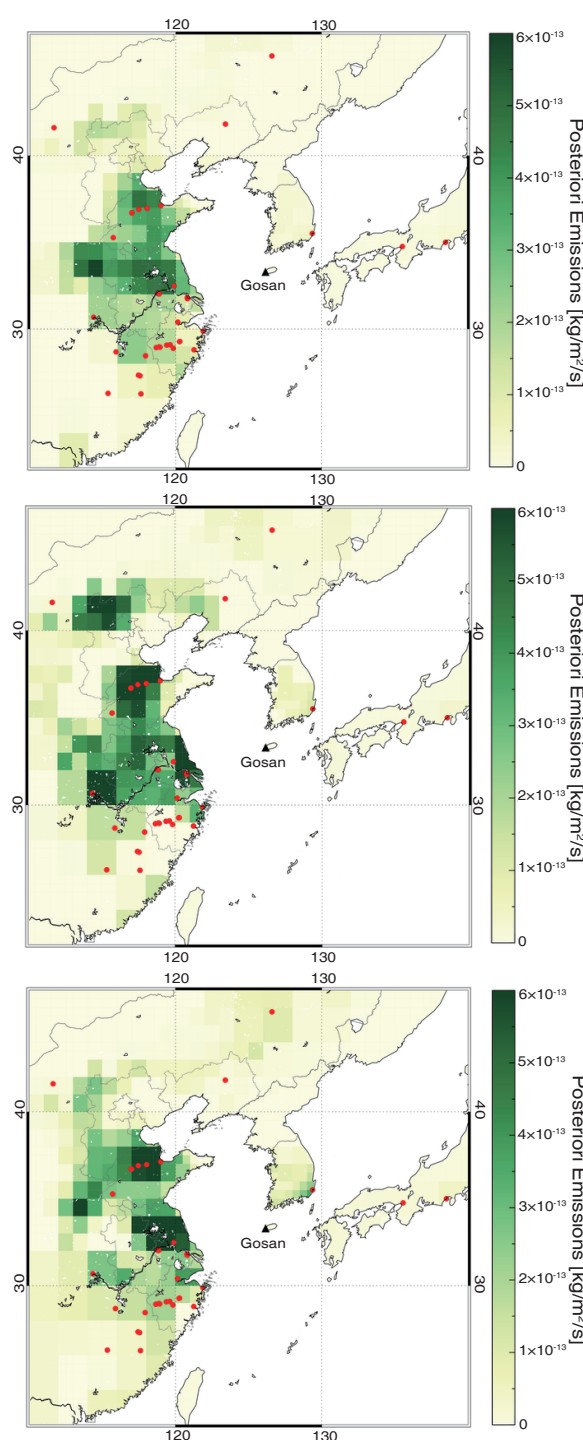

**Figure 3: Maps of inferred HFC-23 emissions during 2008−2012 (top, CDM period), 2013−2014 (middle, end of CDM period), 2015−2019 (bottom, HPPMP period). Red circles indicate locations on known HCFC-22 production plants. The Gosan measurement station is marked with a black triangle.**




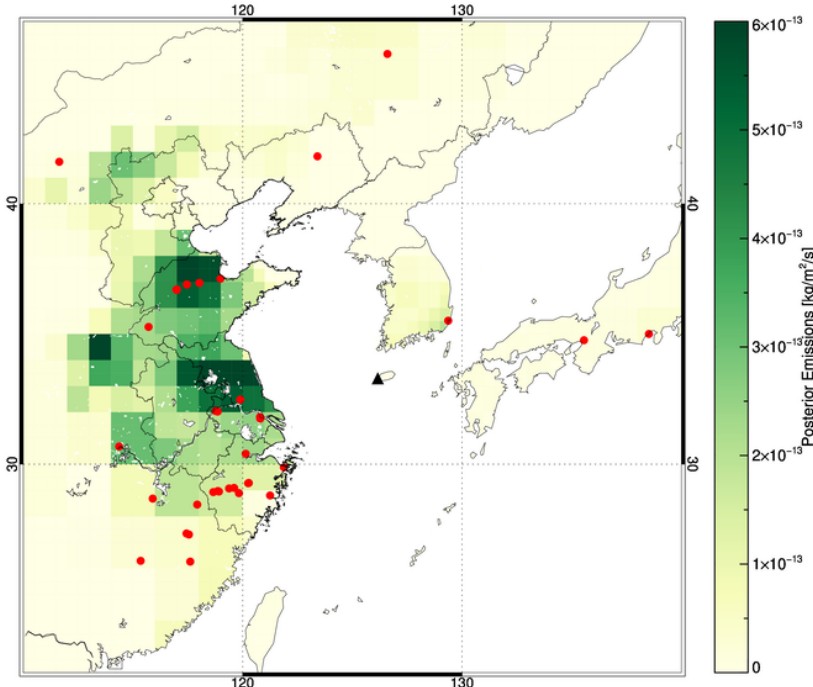

**Figure 4: A posteriori HFC-23 emissions distribution in East Asia for 2008–2019. The red circles indicate locations of known HCFC-22 production plants.**




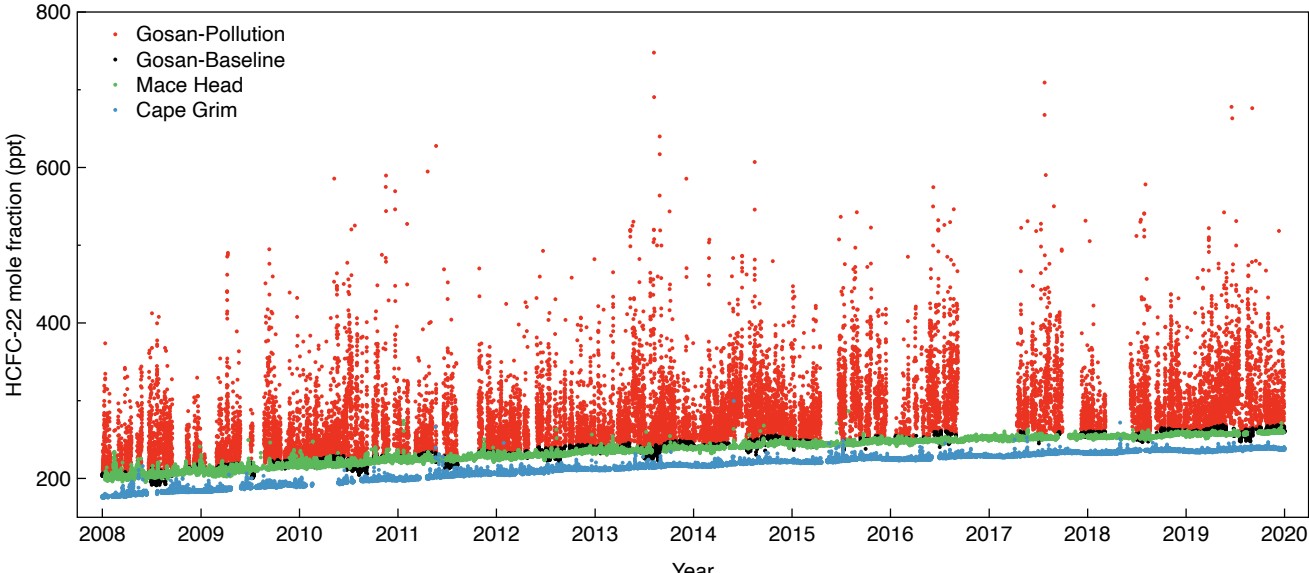

**Figure 5: Atmospheric HCFC-22 concentrations observed from 2008 to 2019 at Gosan. The green and blue dots show HCFC-22 concentrations observed at Mace Head and Cape Grim, respectively, for comparison.**



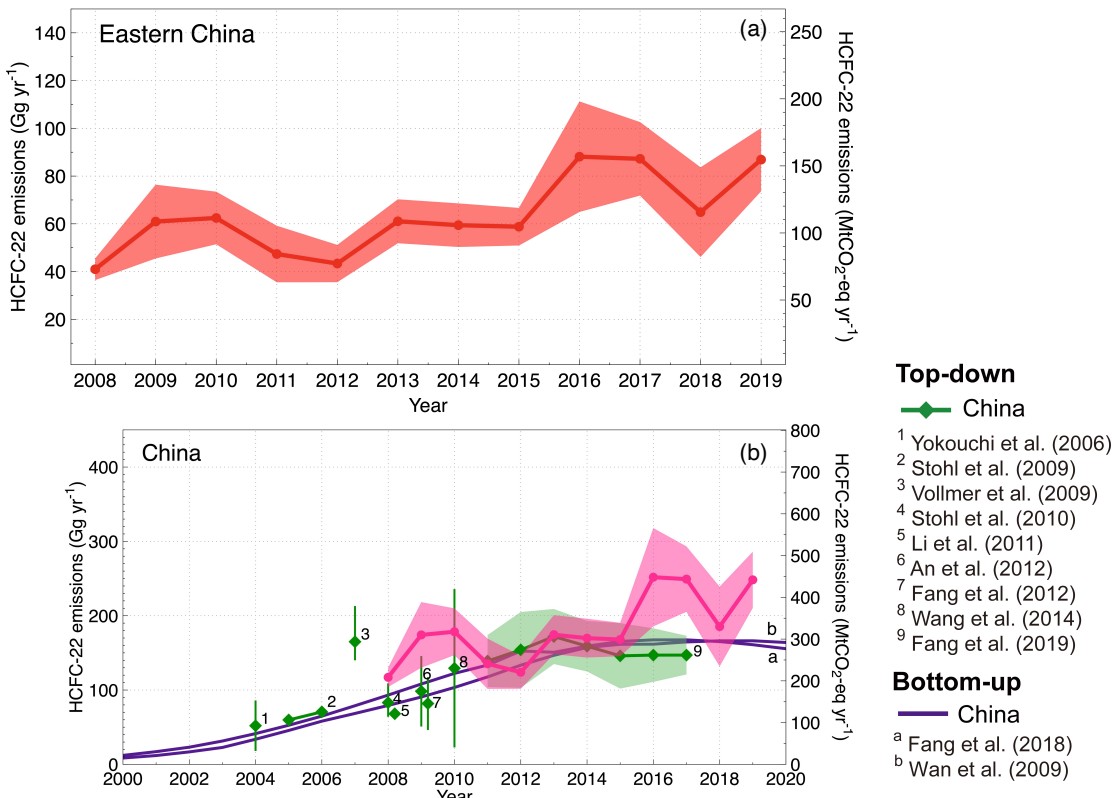

**Figure 6: (a) Eastern Chinese emissions of HCFC-22 (red circles) derived from the atmospheric observations at Gosan, (b) Our**
**HCFC-22 emission estimates by the whole of China (pink circles), determined by scaling up the eastern Chinese emissions by the fraction of the population (35 %) that reside in eastern China (Rigby et al., 2019), top-down Chinese emissions suggested in previous studies (green rhombi), and bottom-up estimates (purple lines).**



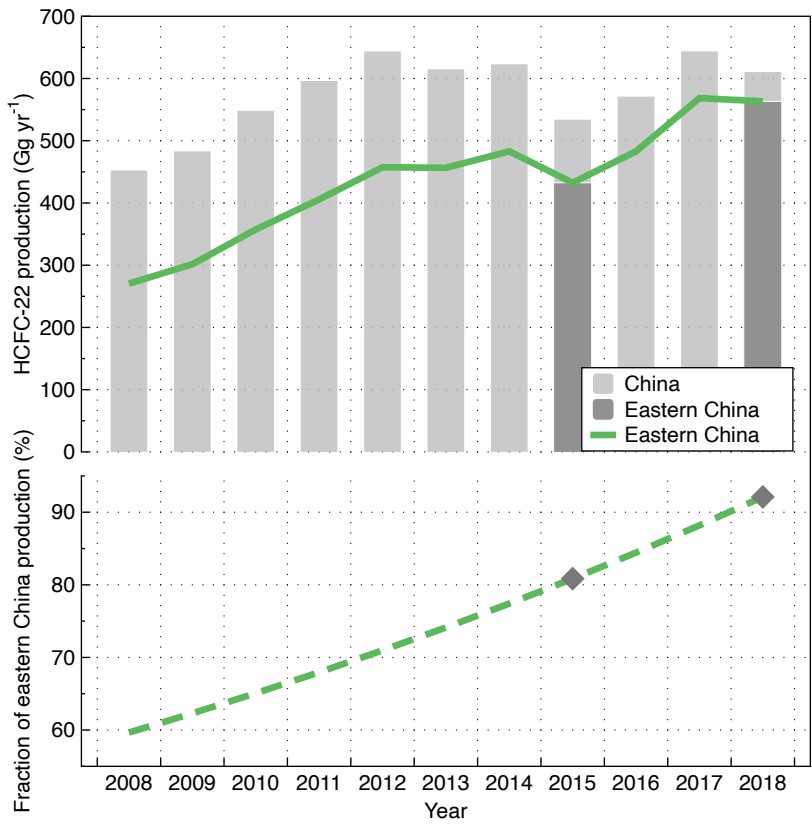

**Figure 7: (a) HCFC-22 productions of China (light gray bars) and of eastern China (dark gray bars) (TEAP, 2021). (b) Interred annual fractions of eastern China HCFC-22 productions against Chinese total productions against Chinese total productions (dark gray rhombi), extrapolated eastern China HCFC-22 production fractions for other years (green dashed line).**



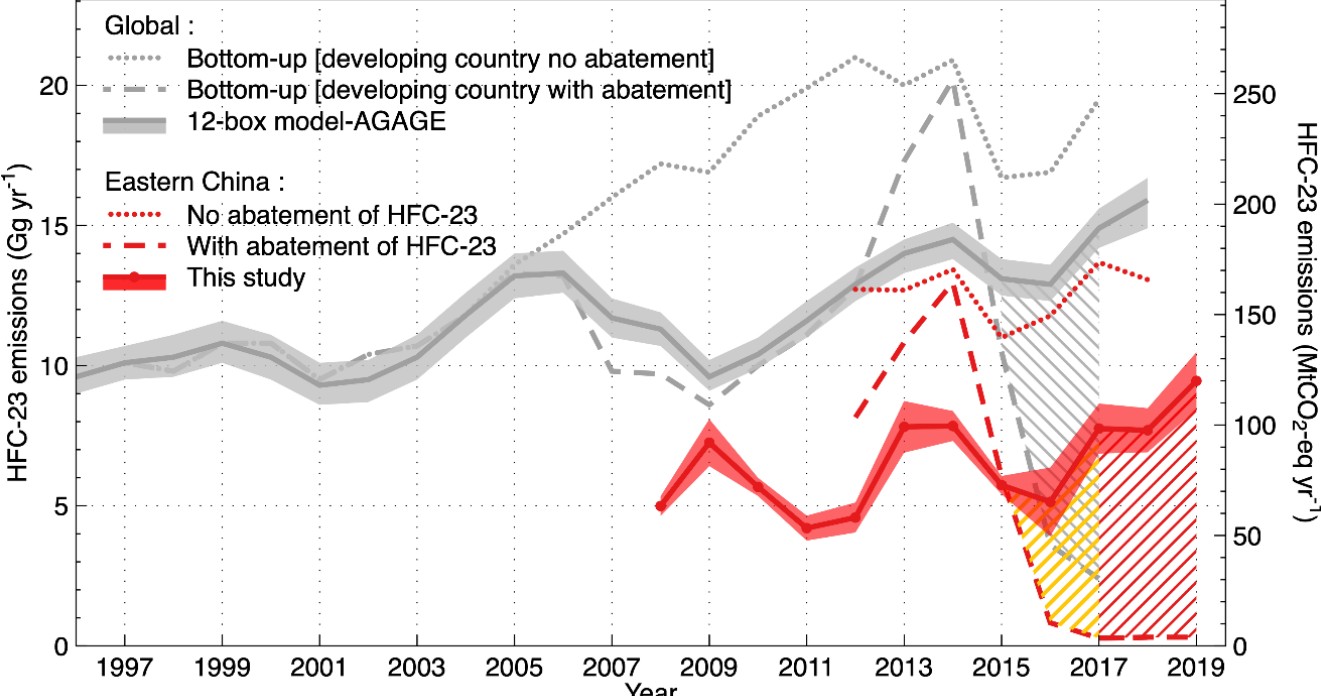


**Figure 8: Observation-based HFC-23 emissions from eastern China versus top-down and bottom-up global HFC-23 emissions derived from AGAGE global data (Simmonds et la., 2018; Stanley et al., 2020). No abatement and abatement bottom-up emissions of eastern China are denoted by red dots line and red dashed line, respectively. No abatement emissions of HFC-23 are determined by HCFC-22 production amounts multiplied by yearly emissions factors of co-produced HFC-23 (TEAP, 2021). HFC-23 emissions**
**estimated with abatement include emissions reduction expected from the CDM monitoring reports (https://cdm.unfccc.int/Projects/registered.html) and the HPPMP (UNEP, 2018).**




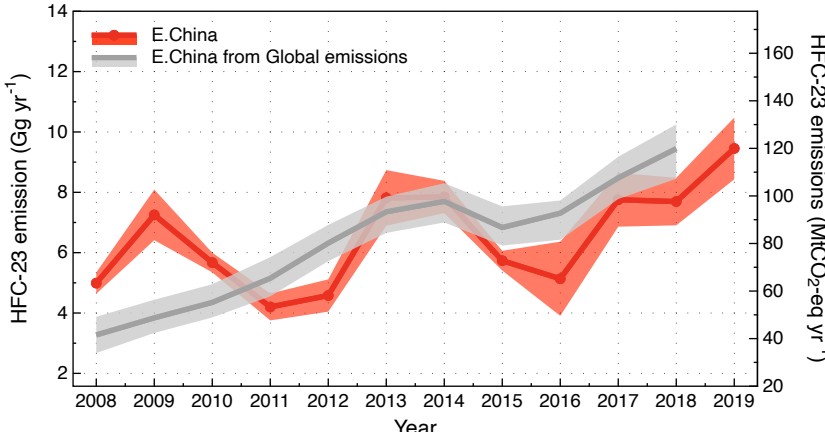

**Figure 9: HFC-23 emissions for eastern China downscaled from the global emission estimates (Stanley et al., 2020) based on the fraction of the eastern China HCFC-22 production (gray line), in a good agreement with our observation-inferred emissions (red line).**



**Table 1: Top-down emissions of HFC-23 for East Asia (eastern China, North Korea, South Korea, western Japan)**

| | HFC-23 (Gg yr$^{-1}$) | | | | | | | | | | | |
|---|---|---|---|---|---|---|---|---|---|---|---|---|
| Year | Eastern China | | | North Korea | | | South Korea | | | Western Japan | | |
| | mean | max | min | mean | max | min | mean | max | min | mean | max | min |
| 2008 | 4.99 | 5.34 | 4.64 | 0.00 | 0.00 | -0.01 | 0.06 | 0.08 | 0.04 | 0.06 | 0.11 | 0.02 |
| 2009 | 7.25 | 8.09 | 6.41 | 0.00 | 0.01 | 0.00 | 0.04 | 0.05 | 0.03 | 0.03 | 0.06 | 0.00 |
| 2010 | 5.67 | 6.00 | 5.35 | 0.01 | 0.01 | 0.00 | 0.19 | 0.24 | 0.14 | 0.05 | 0.08 | 0.02 |
| 2011 | 4.20 | 4.65 | 3.76 | 0.00 | 0.01 | -0.01 | 0.07 | 0.08 | 0.07 | 0.05 | 0.07 | 0.02 |
| 2012 | 4.58 | 5.11 | 4.05 | 0.00 | 0.00 | 0.00 | 0.03 | 0.04 | 0.02 | 0.03 | 0.06 | 0.01 |
| 2013 | 7.81 | 8.74 | 6.89 | 0.00 | 0.01 | 0.00 | 0.10 | 0.12 | 0.07 | 0.07 | 0.10 | 0.03 |
| 2014 | 7.85 | 8.38 | 7.31 | 0.00 | 0.01 | 0.00 | 0.19 | 0.22 | 0.16 | 0.09 | 0.14 | 0.03 |
| 2015 | 5.73 | 6.06 | 5.41 | 0.01 | 0.01 | 0.00 | 0.41 | 0.47 | 0.35 | 0.08 | 0.13 | 0.03 |
| 2016 | 5.14 | 6.36 | 3.91 | -0.01 | 0.01 | -0.02 | 0.16 | 0.23 | 0.09 | 0.10 | 0.17 | 0.04 |
| 2017 | 7.75 | 8.65 | 6.86 | 0.00 | 0.01 | 0.00 | 0.16 | 0.20 | 0.13 | 0.09 | 0.16 | 0.03 |
| 2018 | 7.69 | 8.48 | 6.91 | 0.01 | 0.01 | 0.00 | 0.43 | 0.51 | 0.36 | 0.12 | 0.22 | 0.03 |
| 2019 | 9.45 | 10.48 | 8.43 | 0.00 | 0.01 | -0.01 | 0.34 | 0.40 | 0.28 | 0.21 | 0.39 | 0.04 |