# Peer review of "A rise in HFC-23 emissions from eastern Asia since 2015"

_EGUsphere, 2023_

## Author Comment (AC1)

***Referees' comments on "Continuous increase in East Asia HFC-23 emissions inferred from high-frequency atmospheric observations from 2008 to 2019"***

*by Hyeri Park, Jooil Kim, Sohyeon Geum, Yeaseul Kim, Rona L. Thompson, Jens Mühle, Peter K. Salameh, Christina M. Harth, Kieran M. Stanley, Simon O'Doherty, Paul J. Fraser, Peter G. Simmonds, Paul B. Krummel, Ray F. Weiss, Ronald G. Prinn, Sunyoung Park*

We thank the referees for their thoughtful and thorough reviews. We are pleased that all the reviewers see our manuscript as a valuable contribution to the field. We have taken almost all of the reviewers' suggestions and concerns into account in the revised version, as we note in detail below, and we feel we have been able to make an even stronger presentation of our findings in light of their insights and questions. We thank the reviewers and the editor for their time and effort and appreciate the recommendation for publication in Atmospheric Physics and Chemistry. [In the following, Reviewers' comments are in bold Courier New and our responses and are in Time New Roman font]

**Reviewer comments:**

**Referee #1:**
**Primary comments:**
**This manuscript describes important emission information on key Montreal Protocol controlled and climate related gases. It should be fully published.**
**The methods used to determine the emissions are widely accepted and appropriate for this study.**

**My primary concern is one that can be remedied with some careful attention to the discussion to make the reading of the flow a bit better. The discussion is rather disjointed, with a bunch of factual statements of decisions made for the analysis.**
>>> Based on the reviewer's perspective, we realized that discussions should have been better structured in the previous version without including HCFC-22 observations and emission estimates in the main text. They are now discussed in the Supplementary Information. The HCFC-22 production section of the original manuscript (3.4 Estimation of HCFC-22 production in eastern China) has now been moved to a newly added subsection (2.3 Bottom-up Emissions Estimates) of the Method section to clarify that the HCFC-22 production information for eastern China is linked to the inventory-based bottom-up HFC-23 estimation. We do think the revised manuscript has been improved according to reviewer's suggestions. Thanks for the reviewer's editorial comments!

**There is little discussion of why the decisions were made for many of the assumptions. Such discussion would help to make the logic flow a bit better. One such example is at line 220. An**

**assumption is mentioned with no statement of why. The same is true for the discussion of the results.**

>>> There were three assumptions in the original section 3.4 of the Result and Discussion section. The statement at line 220 in the previous version that "we assumed the eastern China production fractions were correlated exponentially with time", has now been clarified with an addition of the following clause, and is now shown in a new subsection 2.3 of the revised Method section: "because HCFC-22 production rates in developing countries exhibited an exponential growth over time until mid-2010 (https://ozone.unep.org/countries)". The second assumption was that China HCFC-22 plants information reported to UNEP was correct, because we estimated the annual fractions of eastern China HCFC-22 production to the Chinese total based on those reported data. As stated in the text, even if unreported and/or newly built plants exist at unknown locations, our estimation of the abated HFC-23 emissions for eastern China, for which the fractions were primarily used, would not be affected because the emissions abatement action under both the CDM and HPPMP programs must have been taken in the reported facilities. The third one was that a varying rate between the 2015 and 2018 production fractions in eastern China can be applied consistently over time. Given the limited information, the extrapolation approach would not be considered unreasonable.

**All the discussion related to the figures (which I mostly like) does not naturally flow. For example, a better discussion of the comparison of HFC-23 vs. HCFC-22 emissions and what that REALLY tells us would be helpful.**

>>> We have reorganized the figures in the Results and Discussion section by moving the original Figures 5, 6, and 7 (the timeseries plot of atmospheric HCFC-22 observations, observation-derived annual HCFC-22 emissions in eastern China, and annual HCFC-22 production fractions for eastern China, respectively) into the Supplementary Information (Figures S6, S7 and S3, respectively). We think this organization is more concise than our originally submitted version of the manuscript.

The agreement between our top-down HCFC-22 emissions in China and inventory-based bottom-up emissions estimates was taken to suggest that the HCFC-22 production information linked to the bottom-up estimates would also be reasonably accurate, which has been stated both in Section 3.6 and in the completely rewritten Conclusions.

**A useful discussion point that this paper COULD address is the distribution of the inferred emissions relative to what is seen for CFC-11.**

>>> The spatial distribution of the top-down CFC-11 emissions in eastern China (Rigby et al., 2019; Park et al., 2021) showed large emission flux in or around the provinces of Shandong and Hebei for the periods 2014–2017. Given that the emissions of CFC-11 primarily occur during foam blowing, rather than directly during production, the regions where emissions have been identified are not necessarily the same as the locations where the compound has been produced. Whereas the high emission flux densities of HFC-23 were found around the

provinces of Shandong and Jiangsu and were relatively well correlated with the location of known HCFC-22 factories. Considering their distinct emission sources (i.e., CFC-11 use-related emission vs. HCFC-22 production-derived emission), it is unclear whether we could expect a certain correlation between the emission distributions of CFC-11 vs. HFC-23. Also note that it is not possible to discern the location of emissions sources at finer scales than provinces, or groups of provinces in eastern China. Therefore, an explicit discussion of the spatial distribution of HFC-23 emissions in comparison with that of unexpected CFC-11 emissions is beyond the scope of the current manuscript, and thus we have used the annual top-down estimates of CFC-11 only to validate our modeling performance, as in our originally submitted version of the manuscript.

**Other areas that could be described more clearly are how the priors are really constructed relative to the TEAP information regarding the location of the probable production facilities. It was not clear to me how the initial emissions were distributed among the various facilities for the priors, and how they changed.**
>>> We use an ensemble of 27 different inversion runs with a range of prior emissions, which have three different prior flux distributions combined with nine different total prior emission magnitudes with corresponding uncertainties. Detailed information on prior construction is provided in the section of "Three different spatial distributions of a priori emissions" in the Supplementary Information.

As another check, we examined a specific point source distribution for prior emissions based on the likely locations of HCFC-22 production facilities. All plant locations were presumed to have an equal emission flux, which was assumed to be the average of unabated HFC-23 emissions derived from TEAP-based production of known HCFC-22 facilities. For all other areas, 20% of the population density distribution was applied to ensure some minimum prior emissions for all inversion grids. The resulting posterior emissions were consistent with those run by the population density distribution (see the plot appended below).

[Figure]

Figure: HFC-23 emissions from eastern China derived using two different *a priori* distributions: "Population" and "20% of population-based point source" *a prioris*. Each line represents the annual mean of nine different model set-ups for each *a priori* distribution. Shading denotes $2\sigma$ uncertainties.

However, we decided not to include the point source distribution in our inversion analysis based on the following reasoning: (1) potential bias due to inaccuracies in the point source information could be problematic because details on the retirement period of each plant, production magnitudes, and changes in production capacity were not fully available; and (2) applying a point source distribution for prior emissions, which is constructed based on locations of the known HCFC-22 plants, would have already presumed that inferred HFC-23 emissions should be strongly associated with the known HCFC-22 production. Therefore, this approach would not be relevant to investigate the potential existence of unreported, unknown HCFC-22 production.

**A few minor points are below…**
**Abstract: How are the known production locations used in the analysis?**
>>> The production plant locations were compared with the spatial distribution of the inferred emissions in Figure 3 (which was Figure 4 in the originally submitted version). All available information we could find about the facilities was summarized in Table S1 in the Supplementary Information. As noted above, it is unfortunate that we were unable to provide complete information about how long the plants existed, how large their production capacities were/are, and how the production magnitudes have been changed.

**Line 39: the term "basket" is a bit informal. I'd change it to "group"….though did Kyoto really regulate?**
>>> Changed.

**The term "top-down" is loosely defined in the abstract, but not in the body of the paper. I suggest doing so, and with a bit more detail than what is done the abstract.**
>>> We have updated the text in line 23 to read as follows: "A recent study on atmospheric observation-based global HFC-23 emissions (top-down estimates) showed significant discrepancies over 2014–2017 between the increase in the observation-derived emissions and…".

**Figure 3…so that these maps can be shown without the caption, I'd put the years of the analysis in the upper left corner of each map.**
>>> A good suggestion. Updated.

---

## Author Comment (AC2)

*Referees' comments on "Continuous increase in East Asia HFC-23 emissions inferred from high-frequency atmospheric observations from 2008 to 2019"*
*by Hyeri Park, Jooil Kim, Sohyeon Geum, Yeaseul Kim, Rona L. Thompson, Jens Mühle, Peter K. Salameh, Christina M. Harth, Kieran M. Stanley, Simon O'Doherty, Paul J. Fraser, Peter G. Simmonds, Paul B. Krummel, Ray F. Weiss, Ronald G. Prinn, Sunyoung Park*

We thank the referees for their thoughtful and thorough reviews. We are pleased that all the reviewers see our manuscript as a valuable contribution to the field. We have taken almost all of the reviewers' suggestions and concerns into account in the revised version, as we note in detail below, and we feel we have been able to make an even stronger presentation of our findings in light of their insights and questions. We thank the reviewers and the editor for their time and effort and appreciate the recommendation for publication in Atmospheric Physics and Chemistry. [In the following, Reviewers' comments are in bold Courier New and our responses and are in Time New Roman font]

**Referee #2:**

**This manuscript describes new measurements and interpretation that are very important and novel for the international Montreal Protocol community in particular. These results have been eagerly anticipated for some time, given the paper of Stanley et al on global HFC-23 emission changes a few years ago. The current manuscript provides convincing evidence suggesting that the cause of the continued elevated global emissions of HFC-23 arise in large part from eastern China. It will be read and studied extensively by scientists and the much broader audience that includes international policymakers. As such, it needs to be as clear as possible. At this point, I don't have substantive issues with the scientific approach or methodologies, but I do find that the organization and emphasis in certain sections make the main messages opaque and not clearly conveyed. With close attention to these issues in a revision, I believe this paper will be an extremely important contribution to the field.**

>>> Thanks to the reviewer's constructive comments and suggestions. We agree with the reviewer and have completely reorganized the 3. Result and Discussion and 4. Conclusions sections for clarity of presentation. The discussion of HCFC-22 observations and emission estimates is now included in the Supplementary Information. The HCFC-22 production section of the original manuscript has now been moved to a newly added subsection of the Method (2.3 Bottom-up Emissions Estimates) to clarify that the HCFC-22 production information for eastern China is linked to the inventory-based bottom-up HFC-23 emissions estimation.

All the content from the original Conclusions section has now been included in a new subsection titled "3.4 Comparison of HFC-23 emissions in eastern Asia to global HFC-23

emissions" in the Result and Discussion, which now consists of four subsections: 3.1 Atmospheric mole fractions of HFC-23; 3.2 HFC-23 emissions from eastern Asia derived from atmospheric observations using inverse modelling; 3.3 Discrepancy between bottom-up and top-down eastern China estimates; 3.4 Comparison of HFC-23 emissions in eastern Asia to global HFC-23 emissions. We think this organization is much clearer as well as more concise than our originally submitted version of the manuscript.

**For example, the title does not accurately reflect the findings of the paper:**
>>> We agree with the reviewer and have revised the title now to read: "A rise in HFC-23 emissions from eastern Asia since 2015". We think it better reflects our main findings, as the reviewer suggested.

**Emissions from East Asia as a whole are not presented in a table or figure, making it difficult to assess if the sum of emissions from this region have "continuously increased" throughout 2008-2019 as is asserted in the Title (and abstract, see line 26).**
>>> This is a good suggestion, and we have modified Figure 2 and Table 1 to include total emissions from the four regions (i.e., eastern Asia), which must be a useful addition to the paper to show that the observed increase in eastern Asia emissions was primarily driven by emissions from eastern China.
In addition, we have clarified line 26 of the abstract, where we intended to mention the continuous rise for the post-CDM period, by moving "since 2015" to after "the continuous rise".

**Second, the emission estimates for smaller regions (E China, S. Korea, N. Korea, and W. Japan) provided in Figure 2 and Table 1 do not suggest a "continuous increase" in their sum over 2008 to 2019. Instead, I see emissions that vary between 4 and 10 Gg/yr from time to time depending on the effectiveness of the controls in place. Suggestion: revise title to better reflect the results and consider if showing the total from these 4 regions in Figure 2 would be useful.**
>>> As noted above, the revised Figure 2 and Table 1 now show total emissions from eastern Asia. We thought that the term "continuous increase" could also be used to refer to the overall increase from 2008 to 2019, even though it was not a steady increase from year to year. We also intended to indicate the continuing increase in emissions inferred before and after the CDM. However, we now agree with the reviewer that this could be misleading to readers. To avoid ambiguity, we have revised the title to now read: "A rise in HFC-23 emissions from eastern Asia since 2015".

**(note also that East Asia is a well-defined region that doesn't correspond well to the region for which measurements at Gosan have sensitivity to; consider a different label.)**

>>> We thought that "East Asia" would not be a self-explanatory name in a science paper, and thus as far as its detailed definition is provided in the manuscript (see the first paragraph of Section 3.2), it could be used in consistency with many previous studies (e.g., Rigby et al., 2019, Park et al., 2021, Kim et al., 2021, and Western et al., 2022). However, from the reviewer's perspective, we agree that the four sub-divided regions covered by this study are "part of" East Asia, and thus we have re-named our study region as "eastern Asia" to avoid any potential confusion for many readers who are not familiar with our previous studies. So, the term "East Asia" in the original manuscript has now been replaced with "eastern Asia" throughout the text in the revised manuscript.

**The lack of any map indicating the spatial sensitivity extent is unfortunate, particularly since Figures 3 and 4 suggests a strong co-location of HCFC22 production plants with HFC-23 emissions in some instances but not all. Is this because the knowns plants in southern China are not well sensed by measurements at Gosan, or because emissions from these plants were indeed small? A map of surface sensitivity would help address this issue, even if it appeared only in the Supplement.**

>>> A point well-taken. We've now shown the sensitivity of the Gosan observations to the area surrounding the site in Supplementary Figure S1(a) (shown below). We've also indicated the locations of the HCFC-22 production plants that we had identified from various sources (updated Table S1), and boundaries of nine Chinese provinces with high sensitivity in our inverse model domain, to show that six plants located in southern China (Fujian, Jiangxi, and Hubei provinces), four in western China (Sichuan province), one in Inner Mongolia, and one in Heilongjiang, China's northernmost province, are beyond our model domain and thus not included in our emission trend analysis. We note that all other eastern China plants listed in the table were in the high-sensitivity regions where we can potentially retrieve meaningful information through the inversion. It is also the case that further away from the station, atmospheric mixing of plumes that emanate from emissions sources makes their effect on the observations small. Therefore, it was difficult to discern whether the relatively lower strength of HFC-23 emissions from the Zhejiang province suggested actual small emissions or not. So, we include the sensitive map (Figure S1(a)) only to demonstrate that most of HCFC-22 plants were located in the high-sensitivity inverse domain, without adding any speculative explanations.

**(a)**

[Figure]

Figure S1 (a): The sensitivity of the measurements to emissions of HFC-23 (a) for 2008-2019

**Further on figures 3 and 4, some indication of the magnitude of HCFC-22 production by plant (different sized red dots?) would strengthen your arguments suggesting HCFC-22 plants are co-located with HFC-23 emission. Aren't these magnitudes available by production plant at least during the CDM period?**

>>> These are good suggestions, but again it is unfortunate that the details regarding production capacity, operation period, and CDM engagement of each plant were not fully available even for the CDM period because (1) not all HCFC-22 plants in China had participated in the CDM program, and (2) the CDM reports were not submitted on a monthly or yearly basis, making it difficult to accurately estimate HCFC-22 production by plant even during the CDM period. For the post-CDM period, the TEAP reports provided the productions of a few plants in China in 2015 and 201, but the information was not comprehensive and consistent enough to compare production capacities in a quantitative way or to show temporal changes in plant distribution.

**It appears as if the distribution of production plants is the same in all three panels of Figure 3 despite the time changes in HCFC-22 production plants in China through this period. Is it possible to remedy this issue and show only plants reportedly producing during the different periods?**

>>> Considering it is not possible to discern potential time changes in HCFC-22 plant locations, production capacities, and so on, as noted above, we agree that displaying the same distribution of production plants for three different periods can be misleading. Therefore, we decided to mark the identified plants only in the revised Figure 3, which shows the spatial distribution of a posteriori HFC-23 emissions for the entire study period, 2008–2019 (same as the original Figure 4). Whereas, we have removed the HCFC-22 plant locations from the revised Figure 4 (a)-(c) (i.e., previous Figure 3), which are the emission distribution maps grouped for the CDM period, the CDM exit period, and the HPPMP period. The purpose of Figure 4 is to demonstrate

that over the three periods, the broad patterns in the emission sources did not change much, and the emissions during the HPPMP period were similar to or even higher than those for other periods. In addition, a detailed analysis to discern any changes in the location and magnitude of emissions sources over time would not be appropriate given that our emission sources were not identified at finer scales than provinces, or groups of provinces in eastern China. Therefore, the revised Figure 4 does not include the plant locations.

**Discussion related to HCFC-22 is confusing and hard to follow. Some up-front text is needed in sections 3.3. and 3.4 that makes clear why production of HCFC-22 from eastern China is needed with respect to the main points of the paper, HFC-23 emissions, and how you will go about deriving those quantities. Maybe some explanatory text that is in section 3.5 can be moved forward. Consider moving section 3.4 to the Supplement, as it describes a fairly straight forward method for estimating HCFC-22 production in years where that info is lacking.**

>>> As noted in the response to the above comment regarding the organization of the discussion, Sections 3.3. and 3.4 of the originally submitted manuscript have now been moved to the Supplementary Information and Method sections, respectively. A new subsection in the Method (2.3 Bottom-up Emissions Estimates) includes a brief introduction to the inventory-based estimation method originally provided at the beginning of Section 3.5, along with an explanation of why HCFC-22 production in eastern China is needed. Thanks again for the reviewer's editorial comments.

**Conclusion needs a significant overhaul. Issues not discussed in the paper are mentioned at length (unreported HCFC-22 production?) as are potentialities that are in the end discounted. These possibilities can be mentioned, discussed in the paper or Supplement, but they leave this reader very confused when presented only in the conclusion. The Conclusion focus should be on clear statements of the main messages and findings, otherwise they will be lost to most readers, which would be very unfortunate for this most important result.**

>>> We agree with the reviewer. The 4. Conclusions section of the original manuscript has now been reorganized as a sub-section of the 3. Result and Discussion, titled "3.6 Comparison of HFC-23 in eastern Asia to global HFC-23 emissions".

We have completely rewritten the Conclusions of the revised manuscript for clarity of presentation. The revised conclusions now read: "Our top-down inversion estimates for HFC-23 emissions in eastern Asia for 2008–2019 were based on twelve-year, high-frequency in situ observations of atmospheric HFC-23 concentrations made at Gosan. The first published post-CDM HFC-23 emissions in this region show that our observation-derived emissions were much larger than the bottom-up estimates that were expected to be close to zero after 2015 due to

Chinese abatement activities under the HPPMP. Several considerations suggest that observed discrepancies between the top-down and bottom-up HFC-23 emissions estimates are less likely to originate from the existence of unreported, unknown HCFC-22 production. First, the spatial distribution of our derived HFC-23 emissions is well correlated to the locations of the known HCFC-22 plants. Second, our cumulative excess emissions of $23.7 \pm 3.6$ Gg from eastern China between 2015 and 2019 should result in unknown HCFC-22 production of as much as $977 \pm 147$ Gg, equivalent to 136–184 % of the total quantity of HCFC-22 produced for 2018 in eastern China. Such large unknown production seems highly unlikely, given that our observation-based HCFC-22 emissions in China were consistent with the inventory-based HCFC-22 emissions estimates thereby suggesting that the bottom-up emissions for China were relatively well-constrained and thus the associated HCFC-22 production records were also reasonably accurate. Third, when we inferred the eastern China HFC-23 emissions by scaling down the global top-down HFC-23 emissions given in Stanley et al. (2020) based on the fraction of the HCFC-22 production in eastern China to the global total, the HCFC-22 production-scaled HFC-23 emissions were consistent with our observation-derived emissions for eastern China even during the HPPMP period, suggesting that both global and regional HFC-23 emissions, which were independently determined, are primarily associated with known HCFC-22 production facilities. On the basis of this reasoning, the discrepancies between top-down vs. bottom-up emissions of HFC-23 in eastern China are most likely due to unsuccessful reduction processes of HFC-23 at factory level and inaccurate quantification of emission reductions. In addition, our results show that the HFC-23 emissions from eastern Asia explain a substantial fraction of the post-CDM rise in global emissions and are probably due to the unabated emissions in the manufacture of HCFC-22. Thus, they underscore actions to identify responsible industrial practices and organize more effective regulatory controls in order to address this critical issue and prevent unwanted emissions at both regional and global scales. Other regions, and indeed the rest of China, which are not well detected by the current observations may also have contributed to the unexpected trends in HFC-23. Further investigation will be required to better understand which mechanisms are responsible for the recent increase in global HFC-23 emissions in different regions."

**Unclear wording in the abstract:**

**Line 22: on use of the word mandated. The text (lines 55-64) indicates that the Chinese NDRC was a plan for reducing emissions, and the HPPMP is a plan with reporting of emission estimates. None of these comes across as a "mandate", so I'd argue the use of the word is inaccurate unless related laws or mandates were put in place in China that are not currently mentioned in the manuscript.**
>>> Changed to "implemented".

**Line 23- Likely sources of HFC-23 emission are well known (leakage or release of by-production); consider changing "sources of" to "regions responsible for".**

>>> A point well-taken. Done.

**Lines 28-29, are these quantities cumulative for the 5-yr period or annual rates?**

>>> Yes, the quantity represents the 2015-2019 cumulative difference between top-down and bottom-up estimates in eastern China. We have now modified the text (now lines 27-28) here to read: "The cumulative difference between top-down and bottom-up estimates for 2015–2019 in eastern China was ~23.7 ± 3.6 Gg, which accounts for 47 ± 11 % of the global mismatch."

**Lines 30-33. This is a confusing sentence attempting to convey too many points in one sentence. I recommend a rewrite.**

>>> This sentence has been rephrased to (line 29-32): "Our analysis based on HCFC-22 production information suggests the HFC-23 emissions rise in eastern China is more likely associated with known HCFC-22 production facilities rather than the existence of unreported, unknown HCFC-22 production, and thus the observed discrepancy between top-down and bottom-up emissions could be attributed to unsuccessful factory-level HFC-23 abatement and inaccurate quantification of emission reductions."

**Other details:**

**The TEAP data displayed in Figure 7 are puzzling given the expectation that production should have been capped in 2013 and declined thereafter. Is this total production for all uses, dispersive and feedstock-related? Some explanation is needed but lacking.**

>>> The original Figure 7 is now Figure S3 of the revised manuscript. Figure S3 illustrating the TEAP data shows the total production of HCFC-22 in China from 2008 to 2019, including both dispersive and feedstock use. Developing countries agreed to a 10% reduction in HCFC-22 consumption in 2013 and a 10% reduction in production in 2015, followed by a 35% reduction in 2020, a 67.5% reduction in 2025, and a complete freeze in 2030. It is important to note that under the Montreal Protocol, the use of HCFC-22 as a feedstock has been exempted from the phase-out schedule in some countries, including China. According to the TEAP 2020 report, the proportion of China's HCFC-22 production for feedstock use has been increasing relative to HCFC-22 production for dispersive use (a new Figure S2 shown below). Therefore, the lack of a clear decline in HCFC-22 production even after 2013 could be due to an increase in production for feedstock use in China. For this information, the new Figure S2 has been added, and this addition helps to visually represent China's progress toward meeting the targets and clarify the context of the data presented in Figure S3.

[Figure]

**Lines 210-215, without a full analysis of eastern Chinese HCFC-22 results, banks, emissions from banks, etc., it is inappropriate to suggest or "imply that HCFC-22 production and consumption in the region have been increasing" despite MP controls mandating a freeze and decline in these quantities destined for dispersive use in and after 2013. These messages need clarifying.**

>>> A point well-taken. We have changed it: "The continuing rise in the emissions seems to indicate the contribution of dispersive use is still significant, although its production for dispersive use is currently being phased out in developing countries by the Montreal Protocol."

**Citing WMO reports and other assessment reports is appropriately done by using chapter lead authors for the appropriate chapter.**

>>> Changed.

**Lines 49-52, be careful with tense, as Kigali Amendment controls may be "in place" now, but controls on production won't apply for many years for many countries.**

>>> Agreed. This sentence has been rephrased to (lines 52-55): "Later, the HFC-23 by-production and emission associated with the production of HCFC-22 also began to be controlled by the Kigali Amendment to the Montreal Protocol on Substances that Deplete the Ozone Layer (MP), adopted in 2016 (Clark and Wagner, 2016), to preserve the climate benefit achieved by the MP."

**Lines 53-55. Citations appear mid-sentence so support the statement on HCFC-22 production, but the assertion that China is the largest HFC-23 emitter needs support (a citation) given that HFC-23 emission doesn't necessarily track HCFC-22 production given the possibility of destruction.**

>>> We have clarified this (lines 56-58): "The global production of HCFC-22, and thus HFC-

23 emissions, are heavily centered around eastern Asia. Notably, China, where more than 50% of the global HCFC-22 has been produced since 2009, can be considered one of the most significant HFC-23 emitters in the world (UNEP, 2018; Simmonds et al., 2018)"

**Lines 69-72—this argument doesn't hold unless we understand the contributions of these countries, which isn't mentioned writ this point. Logic needs improving.**

>>> We have amended this sentence (lines 68-72) to "A recent top-down study based on atmospheric measurements (Stanley et al., 2020) showed that the global HFC-23 emissions had increased and in 2018 were higher than at any point in history. Stanley et al. (2020) also suggested that these results contradicted the 87 % emission reduction anticipated over 2014–2017 from HFC-23 capture and destruction activities at HCFC-22 production facilities under the national phase-out plans initiated following the end of the CDM programs in a few developing countries including China (UNEP, 2018; Say et al., 2019)."

**Figure 1 and Figure 5, consider somehow indicating periods where the instrument was not operational to prevent the conclusion that there were no measured enhancements during late 2016 to mid-2017 and other times. Also, indicate in the caption if measurements made at Mace Head and Cape Grim represent all results or only those during non-polluted periods (the latter would be most appropriate to show).**

>>> We have revised the caption for clarity to read "Atmospheric HFC-23 concentrations observed from 2008 to 2019 at the Gosan station (GSN, 33.3° N, 126.2° E). Pollution events denoted by red dots were identified as significant enhancements in concentrations from background levels shown in black. Some data gaps in the Gosan measurement timeseries occurred due to instrumental system downtime caused mostly by typhoons and heavy rains. The green and blue dots represent all observations of atmospheric HFC-23 at Mace Head (53.3°N, 9.9°W) and Cape Grim (40.7°S, 144.7°E), respectively, for comparison.

**Figure 2, indicate that the values plotted are annual averages. How were the data gaps handled when deriving annual means and their uncertainties? This needs a clear description in the methods. The uncertainties for 2017 don't look much different than other years, despite there only being approximately only half as many measurements available for interpretation during 2017.**

>>> With regard to the reviewer's question, we would like to clarify that the uncertainties for our inverse modelling results were determined as $2\sigma$ uncertainties of annual means from 27 different model runs, where each set of three different a priori distributions have nine combinations of different a priori emissions magnitudes with corresponding uncertainties. Therefore, the uncertainty estimates account for the variability resulting from different model

configurations and a priori assumptions. The determination of uncertainties for the model results is described in detail in the Supplementary Information.

The measurement error covariance matrix has also been involved in each inversion run (Section 2.2), and the uncertainty assigned to each concentration enhancement incorporates measurement uncertainty and other uncertainties associated with the model representation of this measurement, which are difficult to estimate directly. As such, we performed several inversion tests for 2008 varying the uncertainty and comparing the reduced-chi square value, which is equal to half the value of the cost function divided by the number of observations and served as an approximate measure for the appropriateness of the uncertainties. Then the assigned measurement uncertainties were applied throughout the study period. In this sense, it is possible that overall model uncertainties would vary not only by errors in the meteorology and transport modeling, but also by bias introduced from data gaps. To consider how significant data gaps could be problematic, we have examined the sensitivity of our measurements to HFC-23 emissions for each year between 2008–2019 (see Figure S1(b) shown below). We note that the year-to-year sensitivity of the observations to the emissions from eastern Asia did not change substantially throughout this period, and thus potential bias resulting from the data gap in 2017 would be within the overall uncertainties determined from 27 different model runs.

[Figure]

Figure S1(b): The sensitivity of the measurements to emissions of HFC-23 for each year between 2008–2019.

**Figure 3, axis label units aren't mentioned. Consider including a table next to the panels or within them providing mean emission totals by region.**

>>> Done. The axis unit has been added. Updated Figure 3 and Table 1 now include the total emissions from eastern Asia.

**Figure 7 has typos and duplicate text that needs fixing.**

>>> Corrected.

**Figure 8, hatched regions need defining in the caption, particularly the yellow region, even if they are described in the main text.**

>>> That's a good suggestion. Done.

**Figure 9, assumptions (e.g., emission/production fraction) were made to derive the HFC23 emissions from Chinese HCFC-22 production information shown in this figure, and these assumptions need mentioning in the caption along with a description of the values that were chosen and how they were derived. (Don't make the reader have to search for important details in the main text.)**

>>> As noted in our response to the previous comment, we have downscaled the global top-down emissions of HFC-23 to HCFC-22 production-scaled eastern China HFC-23 emissions using annual fractions of eastern China's HCFC-22 production to the global total production. The annual fractions were derived using the simple extrapolation method described in Section 2.3 and the associated assumptions were also discussed in the text. Since this downscaling itself did not involve any assumptions, we have just expanded the caption of Figure 6 (i.e., Figure 9 in the original version) to read: "HFC-23 emissions for eastern China inferred by downscaling the global top-down emissions estimates (Stanley et al., 2020) based on annual fractions of the eastern China HCFC-22 production to the global total production. The HCFC-22 production-scaled estimates of HFC-23 emissions (gray line) are in good agreement with our observation-based emissions (red line)."

---

## Author Response (AR2)

*Referee #2's comments on "A rise in HFC-23 emissions from eastern Asia since 2015"*
*EGUSPHERE-2023-6*

We thank the reviewers and the editor for their valuable time and efforts in assessing the manuscript. Their recommendation for publication in Atmospheric Physics and Chemistry is sincerely appreciated. We would also like to extend our gratitude to referee #2 for providing us with insightful suggestions and thorough comments on the revised manuscript.

**The authors have made a commendable effort to respond to reviewers' comments and recommendations, and I find the paper to be notably improved. I'd suggest it is acceptable for publication once the following addition points are considered and/or clarified in the subsequent revision.**
>>> We have taken almost all the reviewer's suggestions into account in the rerevised version, as we note in detail below. [In the following, Reviewer's comments are in bold Courier New and our responses and are in Time New Roman font]

**My main issue in the initial version was clarity of writing and presentation. That aspect has been greatly improved in the revision, but there are some issues that would still benefit from further attention.**

**One overarching point to consider: from the Abstract, if HFC-23 is "mainly [not entirely] emitted… as a byproduct in the production of HCFC-22", it seems necessary to acknowledge the possibility that the increase in HFC-23 emission from China after 2015 is related to an industrial process separate from HCFC-22 production and the HPPMP. If the mitigation amounts reported by China in recent years are only provided as percentages of emissions related to HPPMP-covered activities (i.e., not absolute magnitudes, see line 60), then the multiplication of total eastern-Chinese HFC-23 emissions by these factors, as done by the authors, may be inappropriate (please clarify origin of numbers on line 62). While an exploration of the possibility that other processes could contribute HFC-23 emissions outside of the HPPMP after 2014 is outside the scope of this paper, the authors should consider if a caveat on this point should be included with their main conclusions suggesting the failure of the HPPMP.**
>>> Yes, we agree with the reviewer that the HFC-23 emissions related to HCFC-22 by-production would not be the entire HFC-23 emissions, and thus to address the reviewer's comments given here and below, we have clarified throughout the manuscript that our discussion on the rise in HFC-23 emissions from eastern China after 2015 primarily focuses

on the emissions associated with HCFC-22 co-production, following the approach of the previous study on the post-CDM increase in global HFC-23 emissions (Stanley et al., 2020).

The numbers listed on line 62 were calculated based on the reported, annual HCFC-22 productions in China, yearly co-production fractions (%) of HFC-23 to the HCFC-22 production, and reduction percentages of co-produced HFC-23 for 2015, 2016, 2017 and 2018 in the whole of China (listed in Line 60). These reported numbers were in UNEP, 2018 and TEAP, 2021. The reduction percentages for China were reported to UNEP under China's HPPMP.

We have added a few words to lines 62-63 to clarify the above two points "According to China's HPPMP, the *expected* HFC-23 emissions from *HCFC-22 production in* the whole of China should have been 7.5, 1.0, 0.3 and 0.3 Gg yr-1 in 2015, 2016, 2017 and 2018, respectively."

As commented by the reviewer, an exploration of other possible processes that could contribute HFC-23 emissions outside of HCFC-22 production and the HPPMP activity is beyond the scope of this manuscript. Therefore, considering the substantial lack of pre-existing knowledge on such possibilities, mentioning other potential processes in the conclusion that could contribute to the unexpected increase in HFC-23 emissions, apart from HCFC-22 production, would be both vague and potentially distracting.

**On the newly added section 2.3, some clarifications are needed related to the use of the terms HFC-23 emission (amount released to the atmosphere) vs HFC-23 co-production associated with HCFC-22 production (i.e., HFC-23 amounts available for emission depending on the presence or efficiency of mitigation; terms such as "potential emission" seem clearer). Be sure that you intended the first two sentences to refer to HFC-23 emissions, or amounts that were released, and not HFC-23 production (potential emission). Given the apparent misuse (I believe) of the word "emission" in the third sentence, I'm concerned that the authors meant for these first two sentences actually refer to co-production rather than emission. The third sentence mentions unabated emissions, so this refers to co-production or potential emission rather than emission per se.**

>>> A good comment. As explained in the text, the term HFC-23 emissions in the first sentence of section 2.3 are the amount released to the atmosphere, and for non-Annex I countries that do not report national emissions to the UNFCCC, they have been derived from the reported HCFC-22 production multiplied by co-produced HFC-23 percentages (Stanley et al., 2020; Simmonds et al., 2018). As mentioned in the second sentence, they refer to the "expected emissions" when no abatement activities would be applied. However, since we agree with the reviewer that the term potential emissions seem clearer than the expected emissions, we've changed it to potential emissions in line 165.

We've also clarified this point by changing "emission factors of co-produced HFC-23" to "co-production rates of HFC-23" in lines 81, 163, 167, 245 in the text.

**Same point can be made elsewhere in the manuscript, e.g., Figure 5 caption "No abatement emissions of HFC-23 are determined …multiplied by yearly emissions factors of co-produced HFC-23" ? Non-abated emissions aren't derived from "emission factors", aren't they better described instead as being derived from co-production rates?**

>>> The term "emission factors" had been used in previous studies (e.g., Stanley et al., 2020), but as noted in our response to the above comment, the term has now been replaced with "co-production rates" throughout the text and the figure captions in the revised manuscript.

**Line 152, should it be "detrended monthly means"?**

>>> We have modified the text (lines 52-53) for clarify to read "…was determined by a month-to-month extrapolation between one standard deviation values of all the background monthly means similarly to the background concentrations derived at each measurement time over…."

**Line 153, for reference, please provide the average enhancement mole fraction so the reader can understand the relative error associated with variability in the background mole fraction.**

>>> We have added the 2008-2019 average of the enhancement concentrations of 2.92 ppt in line 153.

**Line 175, comment on the implications on your conclusions of deriving HCFC-22 production in years other than 2015 and 2018 with a different approach than expressed by eqn 3 (for example, a constant production fraction over time at the mean of the rates suggested for the reporting years 2015 and 2018). Related to that, the solid green line in the upper panel of Figure S3 needs explaining in the Figure's caption. Also, a spelling issue in the caption: "Inferred" not "Interred" seems likely.**

>>> As suggested by the reviewer, we had tested with a constant production fraction of HCFC-22 over time for eastern China determined from the average of the reported rates in 2015 and 2018 (the first figure shown below), and using the fraction of the HCFC-22 production in eastern China against the global total production, we had calculated the eastern China HFC-23 emissions by scaling down the global top-down HFC-23 emissions. The results in the second figure below showed that the post-2015 HFC-23 emissions were identical within the uncertainty to those determined by the current exponential approach (Eqn. 3, Figure S3 and Figure 6), and the emissions prior to the year 2015 were only slightly higher and thus, became more similar than those for the current approach to our observation-inferred emissions (red line) for eastern China. In light of the similarity between the resulting HFC-23 emissions using exponentially varying vs. time-constant HCFC-22 production fractions for eastern China, we decided not to include and discuss the results from a time-constant fraction in the original

manuscript. This choice was based on the consideration that applying a constant value determined in 2015 and 2018 back in the past appeared less realistic.

[Figure]

>>> The misspelled word has been corrected. Thank you.

**Line 225, It's hard to assess the assertion that emissions in this study are consistent with previous ones for the same years, because Figure S4 shows different quantities (all of China vs the eastern China estimates from this study) and therefore agreement might not be expected (this needs mentioning here) and actually isn't seen for all studies despite different quantities being compared. Also, add to the figure S4 caption an indication of the method used in each study (Atmosphere-based or inventory based).**

>>> A good suggestion. We have added a word "top-down" in the caption of Figure S4 for clarity to read: "…in comparison with previous top-down estimates of….". We have also updated the corresponding text (line 225) by adding the word "top-down".

**Line 273: Figure S5 seems to contain no new information compared to the main Figure 5, so seems unnecessary and confusing as to why it is included.**

>>> Agreed. In the previous revision, we had modified Figure 2 to include total emissions from eastern Asia and to show that the observed increase in eastern Asia emissions was primarily driven by emissions from eastern China. Therefore, we have removed Figure S5 because it is easy to understand that the comparison of the eastern Asia HFC-23 emissions to global emissions is very similar to the case for eastern China shown in Figure 5.

**Line 286: I'm guessing that ~400 Gg/y of HCFC-22 production with unabated HFC23 emission (Emission of HFC23 / production HCFC-22 ratio of ??) during 2015-2019 would be needed to explain the HFC-23 emission excess, not what your wording seems to suggest: that 400 Gg of unreported HCFC-22 actually took place.**

>>> A point well-taken. We have revised the text in line 286 now to read: "Approximately 400 Gg per year of HCFC-22 production with unabated HFC-23 emissions was produced during 2015–2019".

**It would be helpful to indicate if the global estimates in Chapter 2 of the WMO 2022 Ozone Assessment are the same as those shown in the present work (and why not updated to 2020?).**

>>> The global estimates in Chapter 2 of the WMO 2022 Ozone Assessment are identical to those discussed in the main text of this study and shown in Figure 5, except their extension to the year 2020. When we submitted this manuscript in January, the 2022 Ozone Assessment had not been released yet and only the Stanley et al. paper was available, so the updated figures could not be included in this manuscript.